# The Effects of Long-Term Nutrition Counseling According to the Behavioral Modification Stages in Patients with Cardiovascular Disease

**DOI:** 10.3390/nu13020414

**Published:** 2021-01-28

**Authors:** Keiko Matsuzaki, Nobuko Fukushima, Yutaka Saito, Naoya Matsumoto, Mayu Nagaoka, Yousuke Katsuda, Shin-ichiro Miura

**Affiliations:** 1Department of Nutrition, Fukuoka University Nishijin Hospital, Fukuoka 814-8522, Japan; 2Fukuoka Women’s Junior College Health and Nutrition, Fukuoka 818-0193, Japan; fuku.c.fukushima@gmail.com; 3Department of Cardiovascular Medicine, Public Yame General Hospital, Fukuoka 834-0034, Japan; ymhp2388@yamehp.jp; 4Department of Rehabilitation, Fukuoka University Nishijin Hospital, Fukuoka 814-8522, Japan; nmatsumoto@fukuoka-u.ac.jp; 5Department of Pharmacy, Fukuoka University Nishijin Hospital, Fukuoka 814-8522, Japan; mnagaoka@fukuoka-u.ac.jp; 6Department of Cardiovascular Medicine, Fukuoka University Nishijin Hospital, Fukuoka 814-8522, Japan; katsuda@fukuoka-u.ac.jp (Y.K.); miuras@cis.fukuoka-u.ac.jp (S.-i.M.)

**Keywords:** behavioral modification stages, nutrition counseling, patient education

## Abstract

Background: the behavioral modification stages (BMS) are widely used; however, there are no reports on long-term nutrition counseling for cardiovascular disease (CVD) according to BMS. Aim: to study the effects of long-term nutrition counseling based on the BMS in patients with CVD. Methods: fifteen patients with CVD who participated in nutrition counseling were enrolled between June 2012 and December 2016. We provided BMS and dietary questionnaires to estimate the stage score (SS), salt intake, and drinking habits (non-drinking group (*n* = 7)/drinking group (*n* = 8)), and measured the blood pressure (BP), body mass index (BMI), and biochemical markers before and after hospitalization at 6 months, 1 year, and 1.5 years after leaving the outpatient department (OPD). Results: a significant decreased salt intake and increase in SS were found at 1.5 years. It significantly decreased the BP and salt intake in the non-drinking group at 1.5 years. Conclusions: long-term nutrition counseling according to BMS improved salt intake and BP in the non-drinking group. However, in the drinking group, increased salt intake might weaken the BP improvement. Temperance and low-sodium intake are essential factors that control BP, especially in drinkers.

## 1. Introduction

Currently, cardiovascular disease (CVD) is the second leading cause of death along with malignant neoplasm in Japan [1]. In Japan, in particular, where the population aged >65 years exceeds 27.3% of the total population [2], increased mortality from cardiovascular disease has become a social problem. However, the implementation of guideline-based medical therapy in elderly patients with CVD was insufficient to prevent the recurrence of CVD [3].

Lifestyle-related diseases, such as hypertension, are strongly involved in the development of CVD or congestive heart disease (CHF). Therefore, to prevent CVD, daily habits should be improved from early in the day, such as implementing nutritional and exercise therapies and quitting smoking [4,5]. Furthermore, it is reported that recurrence of CHF in elderly patients is caused by their refusal to incorporate behavioral changes—and they continue to consume excessive amounts of salt [6]. Behavioral changes, such as decreased salt intake, are important factors in patients with CVD or CHF. Recently, the trans-theoretical model (TTM) has been used to improve the lifestyle and provide specific guidance to patients with type 2 diabetes [7,8,9,10,11]. TTM theorizes behavioral modification of dietary habits; it suggests behavioral modification by estimating personal readiness to changes and performs specific, individual intervention programs. Behavioral modification stages (BMS) have been indicated in one of the concepts, as well as the preparatory state of psychology, and its practice situations, of five stages: 1) precontemplation (participants have no intention of changing their behavior within the next 6 months); 2) contemplation (they are aware a problem exists and intend to change their behavior within the next 6 months, but not within 1 month); 3) preparation (they are ready for the change within the next 1 month); 4) action (they actively modified their behavior within the last 6 months); and 5) maintenance (they sustained their behavioral changes for >6 months) [12,13].

However, no reports on nutrition counseling using the TTM and long-term follow-up in patients with CVD have been described. We evaluated the effects of long-term nutrition counseling based on the BMS in patients with CVD. Therefore, we hypothesized that long-term nutrition counseling, according to the BMS, should induce lifestyle modification, in particular improvement of high blood pressure and decreased salt intake.

## 2. Materials and Methods 

### 2.1. Investigation Period

This study was conducted from June 2012 to December 2016.

### 2.2. Inclusion Criteria

The participants were patients who had no coronary restenosis after percutaneous coronary intervention at Fukuoka University Nishijin Hospital. The study protocol is indicated in Figure 1. Finally, 15 patients with CVD participated in nutrition counseling. We provided BMS and dietary questionnaires to estimate the stage score (SS), salt intake, and drinking habits, and measured the BP, body mass index (BMI), and biochemical markers at hospitalization to 6 months, 1 year, and 1.5 years after leaving the outpatient department (OPD). Nine, four, and two patients had angina, heart failure (with ischemic heart disease history), and acute myocardial infarction, respectively.

### 2.3. Methods

#### 2.3.1. Research Ethics and Patient Consent

Informed consent was obtained from all patients via consent confirm. The investigation conforms to the principles outlined in the Declaration of Helsinki. The present study was approved by the ethics committee of Fukuoka Medical Association Hospital (Fukuoka University Nishijin Hospital at present) (approval number: N19-03-001) and was prospectively performed form June 2012 to December 2016. 

#### 2.3.2. BMS Questionnaire

Prior to individual nutrition counseling, patients were provided with self-administered BMS questionnaires about dietary intake [8]. Dietary intake in BMS was assessed using five stages of the first described BMS. Furthermore, we performed the scores as follows: precontemplation, 0; contemplation, 1; preparation, 2; action, 3; and maintenance, 4.

#### 2.3.3. Dietary Survey Questionnaire

Patients were provided with three-day food record questionnaires [14], with self-reports regarding their standard dietary intakes (in OPD, 3 days before nutrition counseling) for 3 days before hospitalization. An interview survey on dietary intake was conducted by a managerial dietician during nutrition counseling. To investigate the amount of drinking per day, and the drinking frequency for 1 week, we conducted interview surveys. The calculation of alcohol intake was based on the Standard Tables of Food Composition in Japan, 2015 (Seventh Revised Version) [15]. Furthermore, we calculated the alcohol intake on an average day.

#### 2.3.4. Nutrition Counseling

Nutrition counseling involved making appropriate improvement suggestions in dietary habits and behavior. Cardiologists provided counseling and nutritional intake directions (energy, protein, fat, and salt intake). Counseling to promote a diet focused on increased consumption of, vegetables, mushrooms, seaweed, and fish and decreased consumption of foods that are high in salt (sodium), deep-fried food, and food with added sugar. Recommendations applied to risk factors for CVD, such as high blood pressure and high cholesterol [4,5]. Long-term nutrition counseling included counseling done during hospitalization, and 6 months, 1 year, and then 1.5 years after leaving the OPD. We determined BMS on dietary intake and dietary survey contents, and then nutrition counseling (30-min individual interview) was performed according to the preparatory state (BMS) at each time. The same managerial dietician set a target along BMS with patients at each time point.

Counseling was guided by referring to the Ministry of Health, Labor, and Welfare, Japan Safety, and Health Council (2018): the standard medical checkup and health guidance program (definitive) [10]. There are five stages, as follows.

The precontemplation stage: the managerial dietician confirmed the many benefits of changing dietary habits. We explained that, without changing dietary habits, it is much easier for the recrudescence of CVD.

The contemplation stage: the managerial dietician proposed a behavior change on the dietary habits. High blood pressure and high cholesterol can be improved by increasing consumption of vegetables, mushrooms, seaweed, and fish, while reducing the consumption of food high in salt (sodium), deep-fried food, and added sugars. We explained that such behavior changes could assist in reducing the risk of CVD.

The preparation stage: when patients seemed to have a problem regarding dietary intake, the managerial dietician could assist to establish some behavioral objectives on how to improve. We proposed increasing the consumption of vegetables, mushrooms, seaweed, and fish while reducing consumption of high salt (sodium), deep-fried food, and food with added sugar.

The action stage: the managerial dietician identified the behavior problem then provided appropriate correction actions. We suggested continuing consumption of vegetables, mushrooms, seaweed, and fish, while decreasing consumption of food high in salt (sodium), deep-fried food, and food with added sugar.

The maintenance stage: the managerial dietician identified the plan that prevented interruptions in behavioral modifications, concerning dietary habits with patients. We insisted on the continued increase of consumption of vegetables, mushrooms, seaweed, and fish, and reduced consumption of food high in salt (sodium), deep-fried foods, and food with added sugar. We asked patients with concerns about dietary habits.

We assumed a deviation from and retractability in BMS beforehand and established a target [10,11]. Moreover, we investigated factors such as dining out, usage of prepared food, snacking, and exercise habits.

#### 2.3.5. Dietary Intake

Dietary intake investigated energy, carbohydrate, protein, fat, dietary fiber, potassium, and sodium from a three-day food record questionnaire. The calculation of the salt intake was based on the Microsoft Excel software—Excel Eiyoukun version 8.0 (kenpakusha, Tokyo, Japan) [16] from “Sodium (mg)/1000 × 2.54” [15].

### 2.4. Investigation Item

The data from the medical records were collected—that is, age, weight, BMI (weight (kg)/height (m) × height (m)), total cholesterol (T-Cho), high-density lipoprotein cholesterol (HDL-Cho), low-density lipoprotein cholesterol (LDL-Cho), hemoglobin A1c (HbA1c), uric acid (UA), estimated glomerular filtration rate (eGFR), BP. Regarding alcohol consumption, the patients were divided into the drinking (*n* = 8) and non-drinking groups (*n* = 7). The data on SS, salt intake, and alcohol intake at hospitalization, 6 months, 1 year, and 1.5 years after leaving the OPD, were compared.

The relation between salt intake and each factor (sex, age, snacking, drinking habit) after 1.5 years of leaving was also determined. Regarding alcohol consumption (1.5 years after leaving, the non-drinking group (*n* = 7), the drinking group < thrice a week (*n* = 3), the drinking group > four times a week (*n* = 5)) and salt intake, the drinking habit and the respective compared with salt intake and BP were considered.

### 2.5. Measuring Method of Blood Pressure

We normally inform our patients to come to the hospital at 11:00 a.m., and a clinical nurse measures the patient’s BP. BP was collected using a double cuff electronic sphygmomanometer (Terumo ES-H55).

### 2.6. Calculating Dietary Intake

The calculation of the dietary intake was based on the Microsoft Excel software—Excel Eiyoukun version 8.0 (kenpakusha, Tokyo, Japan) [16], from the standard table of food composition in Japan 2015 (Seventh Revised Version) [15].

### 2.7. Statistical Analysis

All analyses were performed using SPSS Statistics Version 22 (IBM, Tokyo, Japan). The weight, BMI, T-Cho, HDL-Cho, LDL-Cho, neutral lipid, HbA1c, UA, eGFR, energy, carbohydrate, protein, fat, dietary fiber, and potassium, with their means expressed as mean (standard deviation), were compared with salt intake, alcohol intake, BP, SS about dietary intake, drinking habits (the non-drinking group (*n* = 7)/the drinking group (*n* = 8)), and the respective SS and salt intake, BP were performed by statistical analysis of one-way analysis of variance, subjected to the general linear model with replicate.

Using the multiple regression analysis, the relation between salt intake (dependent variable) and each factor (sex, age, snacking, the drinking habit; independent variable) 1.5 years after leaving was identified. Then, a stepwise method was used. Furthermore, multiple comparisons regarding the drinking habits (after 1.5 years, non-drinking group (*n* = 7), drinking group < thrice a week (*n* = 3), drinking group > four times a week (*n* = 5)) and salt intake were done using the Bonferroni correction. The correlation between alcohol intake and salt intake were calculated using the Spearman’s rank correlation coefficient. Student’s t-test was used to between the non-drinking and drinking groups (The weight, BMI, T-Cho, HDL-Cho, LDL-Cho, neutral lipid, HbA1c, UA, and eGFR). Chi-squared test was used for the daily habits, and the medications between the non-drinking and drinking groups. Student’s t-test was used to compare SS and salt intake, BP between the non-drinking and drinking groups, with the two-way analysis of variance (without repetitions). *p* values < 0.05 were considered statistically significant (two-tailed test). 

## 3. Results

### 3.1. Changes in Anthropometry and Blood Biochemical Examination

Table 1 presents the changes in patients’ body. The patients’ respective BMIs shows (mean (SD)): 26.1 (3.2), 25.7 (3.3), 25.7 (3.3), and 26.2 (3.8) kg/m^2^. BMI was significantly lower at 6 months after leaving than that at hospitalization (*p* < 0.05).

### 3.2. Changes in Daily Habits 

Changes in daily habits are indicated in Table 2. Dining out, snacking, prepared food, and exercise habits almost remained unchanged. 

### 3.3. Cardiovascular Disease and Medications

Table 3 presents cardiovascular disease and medication information. Medications for the treatment of diabetes accounted for 11.4%. Alpha-blocker was addition one patient 1 year later about antihypertensive drug. Calcium channel blocker was addition one patient 1.5 years later, and Angiotensin II receptor blocker dose reduction one patient. Antiplatelet aggregation drug come off three patients 1.5 years later. HMG-CoA reductase inhibitor and anticlotting drug were addition respectively one patient 1.5 years later.

### 3.4. Changes in SS Regarding Dietary Intake

Score results of the BMS questionnaire were indicated. Changes in SS regarding dietary intake are indicated in Figure 2a. SS after nutrition counseling were significantly higher than those at hospitalization and 1.5 years after leaving (*p* < 0.01); SS at hospitalization, 6 months, 1 year, and 1.5 years after leaving were 2.7 (0.9), 3.3 (0.7), 3.8 (0.4), and 3.9 (0.3) points, respectively.

### 3.5. Changes in the Salt Intake 

The salt intake in patients at the time of hospitalization and 6 months, 1 year, and 1.5 years after leaving were 10.7 (3.2), 8.7 (1.4), 8.2 (1.4), and 8.3 (1.5) g/day, respectively. It decreased in 1 year as well as 1.5 years after leaving, compared with that at hospitalization (at 1 year *p* = 0.0043, at 1.5 years *p* = 0.032) (Figure 2b).

### 3.6. Changes in the Nutrient Intake

Changes in the nutrient intake are indicated in Table 4. Energy, carbohydrate, protein, fat, dietary fiber, and potassium were almost unchanged. Dietary fiber was of low value over this time period.

### 3.7. Changes in the BP

Systolic BPs (SBPs) of patients at hospitalization and 6 months, 1 year, and 1.5 years after leaving were 134.1 (15.0), 124.9 (17.5), 122.9 (15.3), and 127.0 (19.7) mmHg, respectively (Figure 2c). SBPs significantly decreased at 6 months and 1 year compared with that at hospitalization (at 6 months *p* = 0.013, at 1 year *p* = 0.002); however, an upward trend was seen in 1.5 years from 1 year after leaving. 

### 3.8. Relation between the Salt Intake of Patients 1.5 Years after Leaving and Each Factor

Concerning the relation with salt intake at 1.5 years after leaving, each factor (sex, age, snacking, the drinking habit) was compared. A significant positive correlation coefficient of 0.515 was noted between the salt intake and drinking habit (*p* = 0.049).

### 3.9. Changes in Anthropometry and Blood Biochemical Examination (the Drinking Group and Non-Drinking Group)

Changes in the patient bodies in the non-drinking group were compared with the drinking group, as seen in Table 5. The eGFR of the non-drinking group was significantly lower than that of the drinking group at 1 year and 1.5 years after leaving (at 1 year *p* = 0.023, at 1.5 years *p* = 0.045).

### 3.10. Cardiovascular Disease and Medications (the Drinking Group and Non-Drinking Group)

There was no significant difference between the drinking and non-drinking groups at the time of hospitalization. 

### 3.11. Drinking Habit and SS

SS in the non-drinking (*n* = 7) and drinking (*n* = 8) groups were compared (Figure 3a). In both groups, SS at hospitalization and 6 months, 1 year, and 1.5 years after leaving had significantly increased (the non-drinking group: 2.9 (0.9), 3.4 (0.5), 3.9 (0.4), and 4.0 (0.0) points, respectively; all, *p* < 0.05, the drinking group: 2.6 (0.9), 3.1 (0.8), 3.8 (0.5), and 3.8 (0.5) points, respectively; all, *p* < 0.05). SS of the drinking group tended to be lower than that of the non-drinking group.

### 3.12. Drinking Habit and Salt Intake 

Alcohol intake showed an upward trend in the drinking group at the time of hospitalization and 6 months, 1 year, and 1.5 years after leaving were 20.3 (26.7), 19.6 (27.1), 14.9 (20.3), and 24.6 (31.1) g/day, respectively (*p* = 0.254). At 1.5 years after leaving, salt intake in the non-drinking group (*n* = 7), drinking group < thrice a week (*n* = 3), and drinking group > four times a week (*n* = 5) were 7.5 (0.7), 8.0 (0.6), and 9.7 (1.8) g/day, respectively. The non-drinking group had significantly lower salt intake than the drinking group >four times a week (*p* = 0.041). A significant positive correlation coefficient of 0.628 was noted between alcohol intake and salt intake (*p* = 0.012). Salt intake in the non-drinking (*n* = 7) and drinking (*n* = 8) groups were compared (Figure 3b). At hospitalization and 6 months and 1 year after leaving, in the drinking group, it tended to decrease (11.4 (0.2), 9.2 (0.1), and 8.9 (0.1) g/day, respectively). However, 1.5 years after leaving, it was 9.1 (0.1) g/day, which notably increased (not significant). Salt intake in the non-drinking group, at hospitalization and 6 months, 1 year, and 1.5 years after leaving, were 10.0 (0.2), 8.1 (0.1), 7.4 (0.1), and 7.5 (0.0) g/day, respectively. Furthermore, it significantly decreased at 1 year and 1.5 years compared with hospitalization (*p* < 0.05). From hospitalization to 1.5 years after leaving, salt intake was lower in the non-drinking group than that in the drinking group (*p* < 0.01).

### 3.13. Changes in the Drinking Habit and BP

Changes in the drinking habit and BP are indicated in Figure 3c. A comparison of these between the non-drinking (*n* = 7) and drinking (*n* = 8) groups was determined. In the drinking group, SBPs tended to decrease at hospitalization, 6 months, and 1 year after leaving: 134.1 (11.6), 131.6 (17.0), and 124.4 (11.2) mmHg, respectively (hospitalization vs. 1 year after leaving, *p* < 0.05). However, 1.5 years after, it was 133.8 (17.4) mmHg, which was higher than that in 1 year after leaving (*p* < 0.05).

In the non-drinking group, SBPs significantly decreased at hospitalization and 6 months, 1 year, and 1.5 years after leaving: 134.1 (17.3), 117.1 (13.0), 121.3 (17.9), and 119.3 (17.9) mmHg, respectively (*p* < 0.05). In the non-drinking group, DBPs at hospitalization and 6 months, 1 year, and 1.5 years after leaving had significantly decreased (81.0 (18.0), 65.0 (9.2), 65.4 (13.3), and 62.7 (17.2) mmHg, respectively; all, *p* < 0.05).

## 4. Discussion

This is the first report assessing the effects of long-term nutrition counseling according to behavioral modification stages for patients with CVD. We discussed on what kind of help we can assist. The decision-making and execution power are all left to the patient’s own initiative, and medical staff have no way to interfere. However, diet therapy can allow patients to be treated in their daily lives. In conjunction with changes in behavior, the long-term nutrition counseling is reflected in the reduction of salt intake and the improvement of blood pressure.

This strategy improved long-term behavioral modification, salt intake, and hypertension, and its effectiveness was significant in non-drinking patients with CVD. It is well known that excessive salt intake increases BP [17,18,19,20]. In this study, a significant long-term decrease in salt intake and SBP was observed in the non-drinking group, but not in the drinking-group. Thus, long-term decreased salt intake might have resulted in decreased BP in the non-drinking group. Studies have shown that single or repetitive nutrition counseling alone could lead to decreased salt intake and BP, but the long-term effects were unknown [21,22]. In this study, effective behavioral changed and long-term BP reduction could be obtained by repeating nutrition counseling based on TTM. Thus, it was suggested that behavioral change was an important factor to keep long-term BP reduction.

In the drinking group, SBPs had significantly decreased in the short term, but the effects of BP disappeared after 1.5 years. Long-term nutrition counseling with behavioral modification stages went wrong in the drinking group. It is necessary to grope the planning of temperance and abstinence as a target of behavioral modification in social concerns. Yoshimura et al. reported that alcohol consumption is associated with salt intake and BP, and salt sensitivity improved by reducing alcohol intake [23]. Salt intake was also excessive for the drinking group at hospitalization in this study. Salt intake and SBPs had significantly decreased after 1 year in the drinking group. However, the salt intake and BP tended to increase 1.5 years after hospitalization in the drinking group. Moreover, in this study, 1.5 years after hospitalization, due to intake, beef jerky, salted nuts, and pickle (salt intake >2 g) consumption increased in the drinking group (data not shown) [15]. These were regarded as factors of the salt intake and BP tended to increase. Thus, in the long-term drinkers, increased salt intake during alcohol intake might weaken the effect for BP. Therefore, temperance and stricter sodium reduction [24,25] in drinkers [20,26,27] are essential.

Most of the patients in this study were elderly with an average age of 71 years old. Akita et al. reported that guideline-based drug therapy alone for elderly cardiovascular patients was insufficient to prevent recurrence of CHF [3], and Tsuchihashi et al. also reported that one-third of the triggers for relapse of CHF in elderly patients are because of excess salt intake [6]. Therefore, repetitive nutrition counseling based on TTM was effective in preventing recurrence in elderly patients with CVD. Furthermore, there is no significant improvement about lipid profile and dietary fiber intake. It will be problematic in target setting of nutrition counseling in the future. In this study, target setting to the drinkers of temperance, and stricter sodium reduction, with behavioral modification stages are important. To prevent the development or recurrence of CVD, we should also support the acquisition of self-management skills [28], as well as long-term improvement of dietary intake and daily lifestyle habits. This is crucial because absence of effective behavior changes may exacerbate the symptoms, resulting in long-term consequences. 

The findings of this study have to be seen in light of some limitations. The small sample of patients with CVD. Since this study is small, a large-scale, prospective study is necessary to obtain a final conclusion. Moreover, we were not considering the change of medication in this study, but that was a number of minor changes. 

In conclusion, this study suggests that acquisition of effective behavior modifications by long-term nutrition counseling, according to behavioral modification stages, is important for patients with CVD.

## Figures and Tables

**Figure 1 nutrients-13-00414-f001:**
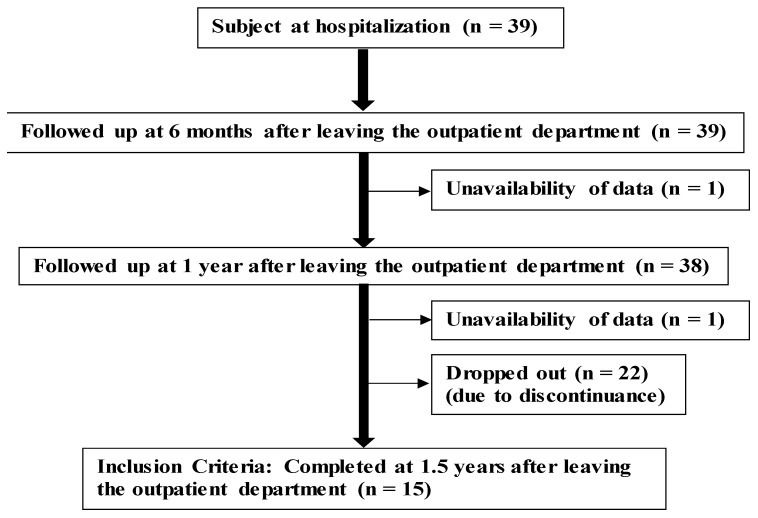
Study protocol.

**Figure 2 nutrients-13-00414-f002:**
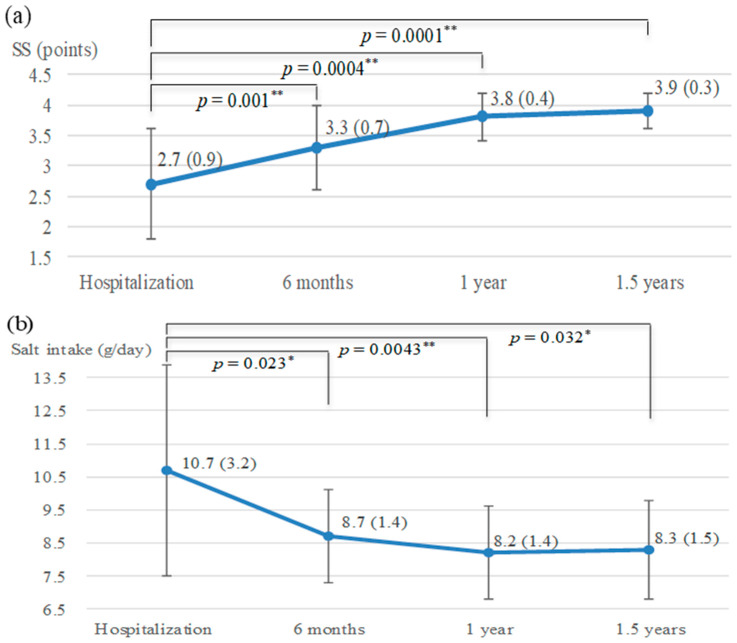
Change in SS regarding dietary intake, salt intake, and BP. Mean (SD). * *p* < 0.05, ** *p* < 0.01, *n* = 15. Abbreviations: SS, stage scores; SBP, systolic blood pressure; n.s., not significant. (**a**) A significant increase in SS was found 1.5 years after leaving compared with that at hospitalization (*p* < 0.01). (**b**) ESI decreased in 1.5 years after leaving compared with that at hospitalization (*p* < 0.05). (**c**) SBPs significantly decreased at 6 months and 1 year compared with hospitalization (*p* < 0.05), however an upward trend in 1.5 years from 1 year after leaving.

**Figure 3 nutrients-13-00414-f003:**
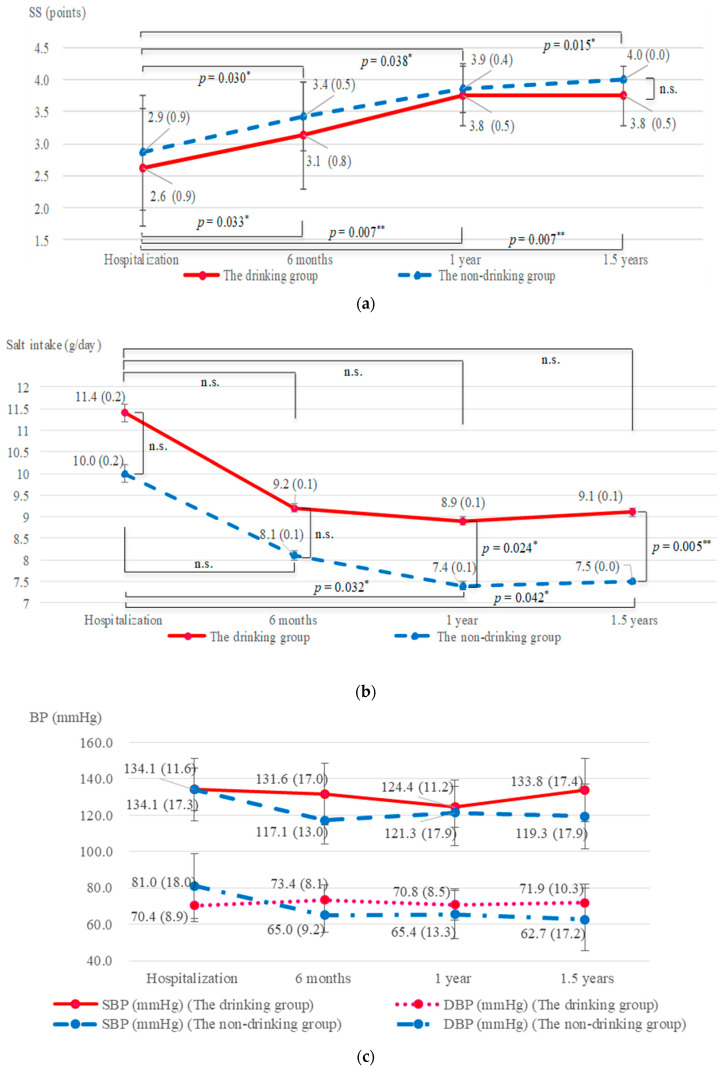
Change in the drinking habit and SS, salt intake and BP. (**a**) In both groups, SS at hospitalization and 1.5 years after leaving had significantly increased (*p* < 0.05). SS of the drinking group tended to be lower than that of the non-drinking group. (**b**) The salt intake in the non-drinking group, 1 year, and 1.5 years after leaving were lower than hospitalization (*p* < 0.05). From hospitalization to 1.5 years after leaving, salt intake was lower in the non-drinking group than that in the drinking group (*p* < 0.01). (**c**) In the non-drinking group, SBPs at hospitalization and 1.5 years after leaving, which showed a decrease (*p* < 0.05).

**Table 1 nutrients-13-00414-t001:** Changes in anthropometry and blood biochemical examination (*n* = 15).

Hospitalization		6 Months	1 Year	1.5 Years	*p* Value		
Male	*n* = 11				Hospitalization vs		
Female	*n* = 4						
Age (years old)	71.3 (8.4)				6 months	1 year	1.5 years
HT (%)	93.3						
DM (%)	53.3						
HL (%)	80.0						
Weight (kg)	69.8 (14.9)	68.7 (14.8)	68.8 (14.8)	70.0 (15.9)	n.s.	n.s.	n.s.
BMI (kg/m^2^)	26.1 (3.2)	25.7 (3.3)	25.7 (3.3)	26.2 (3.8)	0.040 *	n.s.	n.s.
T-Cho (mg/dl)	169.8 (37.0)	174.3 (32.2)	171.8 (31.6)	160.4 (30.2)	n.s.	n.s.	n.s.
HDL-Cho (mg/dl)	50.0 (11.1)	50.6 (13.2)	47.6 (10.3)	47.3 (10.5)	n.s.	n.s.	n.s.
LDL-Cho (mg/dl)	93.1 (34.1)	95.7 (29.6)	94.6 (29.3)	85.3 (29.1)	n.s.	n.s.	n.s.
Neutral lipid (mg/dl)	133.8 (65.4)	140.3 (54.8)	148.1 (71.3)	139.1 (53.2)	n.s.	n.s.	n.s.
HbA1c (%)	6.6 (1.1)	6.5 (0.7)	6.5 (0.7)	6.5 (0.7)	n.s.	n.s.	n.s.
UA (mg/dl)	6.6 (1.0)	6.7 (1.1)	6.6 (0.9)	6.6 (0.8)	n.s.	n.s.	n.s.
eGFR (ml/min/1.73m^2^)	55.8 (17.6)	56.2 (18.8)	53.7 (20.8)	53.8 (21.5)	n.s.	n.s.	n.s.

Mean (SD) * *p* < 0. Abbreviations: HT, hypertension; DM, diabetes mellitus; HL, hyperlipidemia; BMI, body mass index; T-Cho, total cholesterol; HDL-Cho, high-density lipoprotein cholesterol; LDL-Cho, low-density lipoprotein cholesterol; HbA1c, hemoglobin Alc; UA, uric acid; eGFR, estimated glomerular filtration rate; n.s., not significant. BMI was significantly lower at 6 months after leaving than that at hospitalization (*p* < 0.05).

**Table 2 nutrients-13-00414-t002:** Changes in the daily (*n* = 15).

		Admission Time	1½ Years after Leaving	*p* Value
Dining out habits	Yes	11	12	0.307
	No	4	3	
Usage of prepared food	Yes	12	10	0.494
	No	3	5	
Snacking	Yes	14	13	1.000
	No	1	2	
Exercise habits	Yes	10	13	1.000
	No	5	2	

**Table 3 nutrients-13-00414-t003:** Cardiovascular disease and medications (at the time of hospitalization) (*n* = 15).

Drug	Number	(%)
Antiplatelet aggregation drug	23	21.9
HMG-CoA reductase inhibitor	10	9.5
Beta-blocking drug	9	8.6
Vasodilating drug	9	8.6
Calcium channel blocker	9	8.6
Angiotensin IIreceptor blocker	7	6.7
Diuretic drug	6	5.7
DPP-4 inhibitor	6	5.7
Diuretic antihypertensive drug	5	4.8
Environmental Protection Agency drug	4	3.8
Angiotensin-converting enzyme inhibitor	3	2.9
Non-pudding type selective xanthine oxidase inhibitor	3	2.9
Anticlotting drug	3	2.9
Anti-arrhythmic drug	2	1.9
Sulphonyl urea drug	2	1.9
α—glucosidase inhibitor	1	1.0
Biguanide	1	1.0
Long acting insulin	1	1.0
Fast acting insulin	1	1.0

**Table 4 nutrients-13-00414-t004:** Changes in the nutrient intake (*n* = 15).

Hospitalization		6 Months	1 Year	1.5 Years	*p* Value		
					Hospitalization		
					vs		
					6 Months	1 Year	1.5 years
Energy (kcal)	2042.0 (376.9)	1823.2 (284.3)	1789.7 (285.2)	1828.9 (362.4)	n.s.	n.s.	n.s.
Carbohydrate (g)	315.7 (64.2)	283.4 (52.6)	277.8 (51.1)	277.1 (56.7)	n.s.	n.s.	n.s.
Protein (g)	80.3 (13.3)	72.4 (10.5)	72.0 (10.9)	79.3 (25.2)	n.s.	n.s.	n.s.
Fat (g)	50.9 (17.5)	44.5 (13.4)	43.4 (13.4)	44.8 (17.3)	n.s.	n.s.	n.s.
Dietary fiber (g)	16.7 (3.4)	17.0 (2.7)	17.0 (4.7)	15.2 (4.3)	n.s.	n.s.	n.s.
Potassium (mg)	3094.5 (567.7)	2969.3 (553.7)	2976.8 (756.3)	2914.9 (606.5)	n.s.	n.s.	n.s.

**Table 5 nutrients-13-00414-t005:** Changes in anthropometry and blood biochemical examination (the drinking group and non-drinking group).

	The Drinking Group (*n* = 8)				The Non-Drinking Group (*n* = 7)				The Drinking Group vs			
									The Non-Drinking Group			
									Hospitalization	6 Months	1 Year	1.5 Years
	Hospitalization	6 Months	1 Year	1.5 Years	Hospitalization	6 Months	1 Year	1.5 Years	*p* Value			
Weight (kg)	72.4 (18.9)	70.8 (18.5)	70.8 (18.5)	72.1 (20.0)	66.9 (9.0)	66.3 (10.0)	66.6 (9.9)	67.6 (10.5)	n.s.	n.s.	n.s.	n.s.
BMI (kg/m^2^)	26.4 (3.6)	25.9 (3.6)	25.9 (3.7)	26.3 (4.3)	25.7 (2.8)	25.5 (3.1)	25.6 (3.1)	26.0 (3.5)	n.s.	n.s.	n.s.	n.s.
T-Cho (mg/dl)	174.8 (38.1)	173.6 (39.7)	169.8 (41.6)	149.1 (34.7)	164.1 (37.9)	175.1 (24.0)	174.1 (17.1)	173.3 (19.2)	n.s.	n.s.	n.s.	n.s.
HDL-Cho (mg/dl)	52.0 (10.3)	50.9 (13.2)	46.9 (8.4)	49.0 (8.4)	47.6 (12.4)	50.3 (14.3)	48.3 (12.8)	45.3 (12.9)	n.s.	n.s.	n.s.	n.s.
LDL-Cho (mg/dl)	98.7 (41.0)	95.6 (39.4)	95.1 (40.2)	75.3 (36.6)	86.7 (26.3)	95.7 (15.2)	94.0 (11.0)	96.7 (11.2)	n.s.	n.s.	n.s.	n.s.
Neutral lipid (mg/dl)	120.3 (23.5)	135.5 (48.3)	138.8 (63.2)	123.9 (33.8)	149.3 (93.9)	145.7 (65.1)	158.9 (83.4)	156.4 (68.0)	n.s.	n.s.	n.s.	n.s.
HbA1c (%)	6.5 (0.8)	6.5 (0.8)	6.5 (0.8)	6.4 (0.5)	6.7 (1.5)	6.5 (0.7)	6.5 (0.7)	6.6 (0.8)	n.s.	n.s.	n.s.	n.s.
UA (mg/dl)	6.5 (1.1)	6.3 (0.9)	6.4 (1.1)	6.7 (0.9)	6.7 (0.9)	7.2 (1.1)	6.8 (0.4)	6.5 (0.7)	n.s.	n.s.	n.s.	n.s.
eGFR (ml/min/1.73m^2^)	62.8 (13.9)	63.3 (13.6)	64.6 (19.9)	63.9 (20.7)	48.0 (19.1)	48.2 (21.6)	41.2 (14.3)	42.1 (16.8)	n.s.	n.s.	0.023 *	0.045 *

Mean (SD) * *p* < 0.05 Abbreviations: BMI, body mass index; T-Cho, total cholesterol; HDL-Cho, high-density lipoprotein cholesterol; LDL-Cho, low-density lipoprotein cholesterol; HbA1c, hemoglobin Alc; UA, uric acid; eGFR, estimated glomerular filtration rate; n.s., not significant.

## Data Availability

The data that support the findings of this study are available from the corresponding author upon reasonable request.

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
