# Peer review of "The Effects of Long-Term Nutrition Counseling According to the Behavioral Modification Stages in Patients with Cardiovascular Disease"

_nutrients, 2021, doi:10.3390/nu13020414_

Round 1

Reviewer 1 Report

The design of long-term nutrition counseling would be very interesting if the Authors explain more precisely the meaning and importance of behavioral modification stages, and the aim of presented research. It seems difficult to find novelty in the paper without the description of the nutritional/dietetic status of the studied groups. 

Reviewer objections:

  1. lack of information about changes in the patients' dietary habits within 1.5 year observation period,
  2. the main effect was observed in the group of non-drinking patients with CVD in long-term decrease in salt intake and the systolic blood pressure, which is well known from other previous studies, and usually recommended in routine family doctor practice now,
  3. lack of lipid profile changes within the period of 1.5 years is quite astonishing, and short comment on this is necessary,
  4. Table 5 contains the data from Table 3 and Table 4, it may be suggested to remove the last two, 
  5. Discussion should be more suitable and connected with behavioral expectations/recommendations for patients,
  6. Limitations of the study are the small groups and final conclusion.

There is the question about the social and specific to society aspect of the problem presented in the project. 

Reviewer 2 Report

In this manuscript the authors report on the impact of long-term nutrition counseling based on behavioral modification stages in individuals with CVD.

The manuscript needs English language usage editing. There are many sentences that do not make sense as they are written

Line 27: define ESI

Line 35: CVD should be defined on line 33

Line 40: Early stage of what?

Line 42: changes…and they continue to consume excessive amounts of salt and…

Line 43: delete ‘the acquisition’

Line 43: Behavioral changes such as decreased….

The authors need to do a better job at explaining TTM and BMS in the introduction as it is the main point of the manuscript. For example, what are the five stages? Please assume the reader has no knowledge in this topic.

Line 49-51: The sentence is completely disconnected from the previous thought.

Line 53-55: objective should come before the hypothesis. The hypothesis sentence needs to be reworded.

The authors need to do a better job of explaining their methodology. It is very vague as is.

How were the participants recruited?

It is really rare not to see dropouts in studies of this length. Did the study start with 15 participants? Were participants lost to follow-up?

Why does Table 3 only include 8 subjects?

Were the questionnaires used validated? Please provide references.

What was the nutrition counseling? How did it differ at each time point? How did it differ according to BMS stages at each time point?

The authors need to better explain how salt intake was estimated.

Add manufacturer information for instruments used to collect data such as BP.

Was the diet composition calculated? If not, please do so and provide results and information on the software used to do so.

Was data on exercise habits collected?

Line 153: the sentence is repeated

What were the results of the BMS questionnaire?

Table 2: You should also present absolute numbers.

What happened to the medication usage at the other time points?

Line 199-207, 214-219: Provide specific p-values

You need at least four sentences to make a paragraph.

The discussion needs to be considerably improved.

Line 397??

Line 404: Sentence needs to be reworded

Round 2

Reviewer 2 Report

The manuscript is slightly improved, but it still needs significant improvement.

English Language usage is a MAJOR ISSUE.

Line 43: delete 'such as excessive intake of salt or'

Line 100: what type of questionnaire was that? A 24-h recall? A three-day food record? A food frequency questionnaire?

Line 126: aberration??

Line 132: estimated salt intake (ESI) 

Line 133: not sure the timeline makes sense

Line 138: delete 'on a consent form'

2.3.5 should be 2.3.1

Line 157: BP was collected using a double cuff electronic sphygmomanometer. Manufacture information should come in parenthesis. 

The authors did not address the following comments appropriately.

What was the nutrition counseling? How did it differ at each time point? How did it differ according to BMS stages at each time point? They added some information but far from enough.

Was the diet composition calculated? If not, please do so and provide results and information on the software used to do so. You need to provide data on macronutrients, energy intake, dietary fiber, potassium. If you are able to calculate sodium intake, you should be able to calculate other nutrients too.

The authors need to better explain how salt intake was estimated. Just saying that was calculated on excel is not enough.
